# Gastric Microbiota Gender Differences in Subjects with Healthy Stomachs and Autoimmune Atrophic Gastritis

**DOI:** 10.3390/microorganisms11081938

**Published:** 2023-07-29

**Authors:** Giulia Pivetta, Ludovica Dottori, Federico Fontana, Sophia Cingolani, Irene Ligato, Emanuele Dilaghi, Christian Milani, Marco Ventura, Marina Borro, Gianluca Esposito, Bruno Annibale, Edith Lahner

**Affiliations:** 1Department of Medical-Surgical Sciences and Translational Medicine, Sant’Andrea Hospital, Sapienza University of Rome, 00189 Rome, Italy; giulia_pivetta@libero.it (G.P.); ludovica.dottori@uniroma1.it (L.D.); sophia.cingolani@uniroma1.it (S.C.); irene.ligato@uniroma1.it (I.L.); emanuele.dilaghi@uniroma1.it (E.D.); gianluca.esposito@uniroma1.it (G.E.); bruno.annibale@uniroma1.it (B.A.); 2Laboratory of Probiogenomics, Microbiome Research Hub, Department Chemistry, Life Sciences and Environmental Sustainability, University of Parma, 43124 Parma, Italy; federico.fontana1@unipr.it (F.F.); christian.milani@unipr.it (C.M.); marco.ventura@unipr.it (M.V.); 3Department of Clinical and Molecular Medicine, University Sapienza, 00189 Rome, Italy; marina.borro@uniroma1.it

**Keywords:** atrophic autoimmune gastritis, gastric microbiota, gender, hypochlorhydria, healthy stomach

## Abstract

Gender differences and microbiota are gaining increasing attention. This study aimed to assess gender differences in gastric bacterial microbiota between subjects with healthy stomachs and those with autoimmune atrophic gastritis. This was a post hoc analysis of 52 subjects undergoing gastroscopy for dyspepsia (57.7% healthy stomach, 42.3% autoimmune atrophic gastritis). Gastric biopsies were obtained for histopathology and genomic DNA extraction. Gastric microbiota were assessed by sequencing the hypervariable regions of the 16SrRNA gene. The bacterial profile at the phylum level was reported as being in relative abundance expressed as 16SrRNA OTUs (>0.5%) and biodiversity calculated as Shannon-diversity index-H. All data were stratified for the female and male gender. Results showed that women with healthy stomachs had a higher gastric bacterial abundance and less microbial diversity compared to men. Likely due to hypochlorhydria and the non-acid intragastric environment, autoimmune atrophic gastritis seems to reset gender differences in gastric bacterial abundance and reduce biodiversity in males, showing a greater extent of dysbiosis in terms of reduced biodiversity in men. Differences between gender on taxa frequency at the phylum and genus level in healthy subjects and autoimmune atrophic gastritis were observed. The impact of these findings on the gender-specific natural history of autoimmune atrophic gastritis remains to be elucidated; in any case, gender differences should deserve attention in gastric microbiota studies.

## 1. Introduction

Autoimmune atrophic gastritis (AAG) is a chronic inflammatory condition that leads to atrophy of the oxyntic mucosa, reduced secretion of gastric acid, and intrinsic factor associated with frequent positivity of gastric autoantibodies against parietal cells and/or intrinsic factor [1]. Also, low pepsinogen I levels and hypergastrinemia are frequently associated with AAG, and for this reason are considered damage-atrophy biomarkers [2]. The exact pathogenesis of AAG is not yet completely understood. It has been reported that the effectors of the autoimmune destruction of oxyntic mucosa are autoreactive T cells, whose trigger remains to be clarified, even if *Helicobacter pylori* has been supposed to eventually play a role due to antigen cross-reactivity [3]. AAG’s first biochemical manifestation is macrocytic (pernicious) or microcytic anemia, due to cobalamin or iron deficiency, respectively. Dyspepsia, epigastric pain, and post-prandial fullness, but also reflux disease, may be present in about half of patients [4]. AAG should be considered a pre-neoplastic condition potentially leading to gastric dysplasia and adenocarcinoma [1,5] and type I neuroendocrine tumors [6,7,8]. AAG is frequently associated with other autoimmune disorders, mainly autoimmune thyroid disease (Hashimoto thyroiditis) and type 1 diabetes [9,10]. 

In these last years, gender and sex-related differences have been gaining increasing attention in different clinical fields, particularly in autoimmune diseases. Gender may play a role in the prevalence, severity, and altogether on the natural history of autoimmune diseases such as AAG. Possible explanations for gender differences observed in some diseases are, among others, gender-specific sex hormones, gender-sex differences in immune response, and typical gender-related organ vulnerability [11,12]. 

Also, the fascinating world of human microbiota has gained a lot of interest in the scientific community. Previous studies showed that AAG may be associated with modifications in the gastric microbiota composition; indeed, hypochlorhydria as a result of oxyntic mucosa atrophy and impaired gastric acid secretion in this condition makes plausible the overgrowth of intragastric bacteria [13,14,15]. Some studies pointed out the potential role of the gut microbiota as a potential influencer of gender predisposition to diseases [16]. To our best knowledge, data on gender differences in gastric microbiota are lacking so far. Therefore, in this study, we aimed to investigate gender differences in the gastric microbiota composition in subjects with healthy stomachs and patients affected by AAG.

## 2. Materials and Methods

This is a post hoc analysis of a recent study on the gastric microbiota composition in patients with corpus atrophic gastritis [13]. Herein, of the originally included patients, the two *H. pylori*-positive patients were excluded ab initio. Therefore, we conducted a cross-sectional study that included a total of 52 adult subjects (median age 61 years, range 18–83, 32 females and 20 males) who underwent gastroscopy in our centre due to the new onset of dyspeptic symptoms such as epigastric pain and postprandial fullness, reflux disease or anaemia. Patients that were taking anti-secretory drugs were told to withdraw at least three weeks before gastroscopy. All patients with a known diagnosis of AAG were off anti-secretory drugs. Also, antibiotics were withdrawn at least three weeks before the gastroscopy. All the included patients were on their habitual Mediterranean diet without specific restrictions [13]. 

Gastric biopsies were obtained for histopathological evaluation according to the updated Sydney System, sampling a total of five biopsies: two from the antrum, two from the corpus, and one from the *incisura angularis* [17]. The histopathological evaluation identified 30 subjects (57.7%) with healthy stomachs and 22 (42.3%) patients affected by *H. pylori*-negative AAG, defined by the atrophic involvement of the only corpus mucosa, sparing the antrum [18,19]. The main clinical characteristics of the included patients are shown in Table 1.

### 2.1. Gastric Biopsy Sampling

As previously reported [13], at least five gastric biopsies were obtained in each patient during gastroscopy and were processed for histopathological evaluation according to the updated Sydney system [17]. In both, subjects with healthy stomachs and AAG cases, in addition to biopsies for histopathological assessment, antral (*n* = 1) and corpus biopsies (*n* = 1) for genomic DNA extraction and 16SrRNA gene sequencing, were obtained at the start of gastroscopy and immediately frozen at −20 °C. The antral and corpus biopsies sent for gastric microbiota assessment were obtained from the greater curvature immediately adjacent to those sent for histopathological assessment [13]. The acquisition of the biopsies used in this study was approved by the University Sapienza ethics committee (*n*° 7022/2020). Informed consent was obtained from each patient included. The study protocol conformed to the ethical guidelines of the 1975 Declaration of Helsinki (6th revision, 2008) as reflected in a priori approval by the institution’s human research committee.

### 2.2. Bacterial Microbiota Identification by 16SrRNA Gene Sequencing and Data Analysis 

Gastric biopsies were used for genomic DNA extraction. As previously reported [13], genomic DNA from gastric mucosa samples was purified using the PureLink™ Microbiome DNA Purification Kit (ThermoFisher Scientific, Waltham, MA, USA), following the manufacturer’s instructions. A DNA library was created using the Ion16S™ Metagenomics Kit (ThermoFisher Scientific, Waltham, MA, USA), including two sets of primers to amplify the corresponding hypervariable regions of the bacterial 16SrDNA gene: primer sets V2-4-8 and V3-6,7-9. Template preparation and chip loading were carried out using an IonS5 Ion Chef™ System, and sequencing was performed using an IonS5 System. After sequencing, the fastq files were processed using a custom script based on the QIIME2 software [13]. Quality control retained sequences with a length between 140 and 400 bp and a mean sequence quality score > 20, while sequences with homopolymers >7 bp and mismatched primers were omitted. To calculate downstream diversity measures (alpha and beta diversity), 16SrRNA Operational Taxonomic Units (OTUs) were defined at 100% sequence homology [13]; OTUs not encompassing at least 2 sequences of the same sample were removed. All reads were classified to the lowest possible taxonomic rank using QIIME2 software and a reference dataset from the SILVA database v.132 [13].

As previously reported [13], the bacterial profiles at the phylum and genus levels were reported as relative abundance and were represented through bar plots; only taxa with relative abundance > 0.5% were shown. Biodiversity within a given sample (alpha-diversity) was represented by box-and-whisker plots, and the Shannon-diversity index-H was computed.

Data were stratified for female and male gender separately in subjects with healthy stomachs and AAG patients.

### 2.3. Statistical Analyses

Statistical analyses were performed with SPSS software v. 25 (www.ibm.com/software/it/analytics/spss/ (accessed on 5 June 2023)) and MedCalc Statistical Software 20.113 (MedCalc Software, Ostend, Belgium; http://www.medcalc.org; 2022 (accessed on 5 June 2023). Descriptive statistics were performed using mean, SD, median, and range for quantitative variables. The frequencies and percentages were computed for qualitative variables. The bacterial profiles at the phylum and genus levels were reported as relative abundance and were represented through bar plots; only taxa with relative abundance > 0.5% were shown. Biodiversity within a given sample (alpha-diversity) was represented by box-and-whisker plots, and the Shannon-diversity index-H was computed. To compare the abundance and diversity of bacterial genera between groups, Mann–Whitney test was applied. To compare biodiversity (Shannon index), Student’s *t*-test was used. Similarities between samples (beta-diversity) in female and male subjects with healthy stomachs and patients with AAG were calculated by Permanova. A Principal Coordinates Analysis (PCoA) 3D representation of beta-diversity of bacterial genera abundance subdivided by gender based on the Bray–Curtis matrix was performed using Origin Pro 2023. In the PCoA, each dot represented a sample that was distributed in tridimensional space according to its bacterial composition. Lda Effective Size (LEfSe) analysis (Galaxy system, https://toolshed.g2.bx.psu.edu (accessed on 3 July 2023) was performed based on the average relative abundance of all OTUs with >50% prevalence in female or male subgroups. Moreover, a Heatmap based on the average relative abundance of all genera that showed a prevalence score > 50% between at least one of the male and female subgroups of samples was generated. A *p*-value of less than 0.05 was considered statistically significant.

## 3. Results

### 3.1. Gastric Bacterial Abundance

The gastric abundance and composition of bacterial genera between subjects with healthy stomachs and AAG patients were confirmed to be significantly different (*p* = 0.005, R2 0.03733). Further, the gastric bacterial abundance and composition between females and males of healthy subjects and autoimmune gastritis patients were significantly different (*p* = 0.014, R2 0.02797), as illustrated in the PCoA 3D representation of the female and male subjects with healthy stomachs and patients with AAG showing a separate clustering of females and males (please see Figure 1).

As shown in Table 2, in subjects with healthy stomachs, women showed a higher gastric bacterial abundance compared to men, reaching statistical significance at all but 23,333 reads. Differently, in patients affected by AAG, gastric bacterial abundance was similar at all reads in both genders.

As shown in Figure 2, at 13,333 reads, OTUs of female subjects with healthy stomachs were 704.6 (284.1–1145.0) vs. 470.4 (159.6–1116.6) of males (*p* = 0.03); OTUs of female patients with AAG were 467.2 (174.6–1042.2) vs. 433.0 (160.6–560.5) in males, *p* = 0.32).

### 3.2. Biodiversity

As shown in Table 3, in subjects with healthy stomachs, the gastric microbiota was significantly less complex, with a lower biodiversity in females compared to that of males (3.2 vs. 3.4, *p* = 0.003). In contrast, in the AAG patients, the gastric microbiota were significantly less complex, with a lower biodiversity in the males than that in females (Shannon index 2.3 vs. 3.5, *p* < 0.0001). Compared to subjects with healthy stomachs, in the AAG patients, the biodiversity of the gastric microbiota was significantly lower in males (2.3 vs. 3.4, *p* < 0.001), while it was higher in females (3.5 vs. 3.2, *p* = 0.002).

### 3.3. Taxa Frequency at the Bacterial Phylum and Genus Level

Appendix A shows the histogram plot of the taxa frequency at the bacterial phylum level in subjects with healthy stomachs and in AAG patients concerning gender.

As shown in Table 4, at the bacterial phylum level, in subjects with healthy stomachs, Proteobacteria, Firmicutes, Bacteroidetes, and Actinobacteria were reported as the most frequent bacterial phyla. Firmicutes and Actinobacteria were more frequent in females compared to males (*p* = 0.0155 and *p* = 0.0042, respectively). In contrast, the frequency of Fusobacteria in subjects with healthy stomachs was similar in males and females (*p* = 0.4634).

Among patients with AAG, the most frequent gastric phyla were Firmicutes, Proteobacteria, and Bacteroidetes, but their frequency was similar in the two genders. Also, Actinobacteria were quietly frequent in AAG patients but not different in males and females. In contrast, Fusobacteria were significantly more frequent in female AAG patients compared to males (*p* = 0.0124).

At the bacterial genus level, in subjects with a healthy stomach, the most frequent gastric bacterial genera were *Streptococcus*, *Prevotella 7*, and *Granulicatella*. Each one of them was mostly represented in females and less so in males (*p* = 0.0358, *p* = 0.0269, and *p* = 0.0020 respectively). Also, *Neisseria* was quietly frequent in subjects with healthy stomachs, but in this case, no statistically significant gender difference between males and females was observed (*p* = 0.4899). 

On the other hand, in the AAG patients, the most frequent gastric bacterial genus was represented by *Streptococcus*, with no statistically significant gender difference between males and females (*p* = 0.7751). Also, *Prevotella 7* was quietly frequent, but even in this case, the frequency in males and females was similar (*p* = 0.4899). In contrast, *Gemella*, *Haemophilus*, and *Neisseria* were significantly more frequent in females than in males: *p* = 0.0057, *p* = 0.0223, and *p* = 0.0156 respectively (see Table 5).

### 3.4. Heatmap and LEfSe Analysis

To visually display the gastric microbiota taxonomy in the subgroups, a heatmap based on the average relative abundance of all genera with a prevalence score > 50% between at least one of the male and female subgroups of samples is shown in Appendix A.

Moreover, LEfSE analysis based on the average relative abundance of all OTUs with >50% prevalence in the female or male subgroups at the genus level was performed. Figure 3 shows the LEfSe bar charts according to their LDA score/effect size, showing a significant differential abundance of gastric microbiota between the female and male patients. In particular, the gastric microbiota of the females was enriched amongst others with Streptococcus, Granulicatella, and Gemella with a significant difference when compared to males, while the stomach of male patients was significantly enriched with other bacteria, for example, U.m. of Burkholderiaceae family and Lactobacillus, thus confirming the differential bacterial abundance between genders. 

## 4. Discussion

To our best knowledge, this is the first study that aims to investigate gastric microbiota gender differences in subjects with healthy stomachs and in patients affected by AAG. Our results showed that the bacterial abundance of the gastric microbiota is different across the female and male genders, and women with healthy stomachs had a nearly two-fold-higher gastric bacterial abundance but reduced biodiversity compared to men. Thus, in healthy women, the gastric microbiota seems to be more abundant, but at the same time less complex than in males.

This observed difference in the gastric microbiota between females and males might be interpreted as gender- and/or sex-related. The stomach is in communication with the external environment through the oesophagus and the oral cavity. Many bacteria found in the gastric microbiota are also found in the oral cavity and join the stomach after swallowing saliva or food [20]. Thus, it is likely that the gastric microbiota composition might be influenced by environmental factors such as food, water, saliva, and eating habits. But, it has also been reported that the modification of the gut microbiota according to the diet may occur due to a possible sex-dependent effect [16], even if data on the gastric microbiota on this issue are not available yet. In this view, the observed higher bacterial abundance and the less complexity and diversity in women compared to men might be explained by the frequently different lifestyles and eating habits between women and men. Women more frequently follow a healthy diet, with a higher intake of whole grains, vegetables, fruits, nuts, and fish that have strong anti-inflammatory power [21]. Another factor that can influence the composition of the gastric microbiota is oral hygiene and tooth health. Changes in periodontal health status are associated with alterations in the bacterial community [22]. Sun et al. showed how periodontal outcomes were highly correlated with the burdens of *P. gingivalis*, *T. forsythia*, and *T. denticola* and how the presence of those microorganisms was a significant predictor for the development of precancerous lesions of gastric cancer [23]. Thus, different gender-related behaviours regarding oral hygiene habits may play a role in oral health, periodontal outcomes, and, as a likely consequence, gastric microbiota. All these factors can influence the gastric microbiota composition and play a role in maintaining a eubiotic intragastric environment, helping to avoid dysbiosis or, on the contrary, induce gastric dysbiosis. On the other hand, it cannot be excluded that the observed gastric microbiota differences between healthy females and males are due to intrinsic characteristics resulting from the genetic makeup of each individual. The specific role of gender- and sex-related factors in increasing bacterial abundance and reducing complexity and diversity in women compared to men remains to be elucidated. 

From a previous Korean study conducted on patients with gastric cancer and healthy controls, it can be extrapolated that females with healthy stomachs had higher biodiversity with a major complexity of gastric microbiota than males, even if the *p*-value was not reported [24]. This contrasting result may probably be explained by different genetic backgrounds, different eating habits, and different social roles of women and men compared to South Europe.

In AAG patients, the gender difference in gastric bacterial abundance seems to be reset, probably due to hypochlorhydria and the resulting non-acid intragastric environment that irretrievably modifies the microbiota composition permitting the survival of non-acid-resistant bacteria. More specifically, in our study, we found out that even if healthy women had a nearly two-fold gastric bacterial abundance compared to men, hypochlorhydria halved the bacterial abundance in women and had only a scarce impact on it in men. This finding might be interpreted as a gender-specific difference, but sex-related factors such as sex-specific immune or inflammatory responses or hormone influences cannot be excluded a priori. This idea is supported by the gender- and sex-related differences found in AAG. A previous paper reported that AAG was preponderant in women who showed stronger autoimmune serological responsiveness and different HLA-DRB1 association. Particularly, HLA-DR4 and HLA-DR2 alleles were reported as more frequent in females than in males. Moreover, AAG showed differential clinical profiles in female and male patients, occurring mainly in normal-weight, dyspeptic women with iron-deficiency anaemia and autoimmune thyroid disease, and in overweight male smokers with pernicious anaemia [25].

In AAG patients, the gastric microbiota of the males was less complex, with a lower diversity than that of the females. Overall, it seems that the chronic inflammation, together with hypochlorhydria in AAG, significantly reduces the biodiversity of the gastric microbiota in males, while increasing it in females, producing an opposite effect in the two genders. 

Gastric acidity plays a role in the bactericidal defensive barrier and has digestive and absorptive properties. The progressive destruction of hydrochloric acid-secreting parietal cells is typical in AAG and leads to hypochlorhydria. The pH increase due to the weakening of the gastric acid defensive barrier results in consequent gastric microbiota composition alterations with potential overgrowth of bacteria other than *Helicobacter pylori*, known as the main gastric bacterial colonizer [13,14].

Our findings showed that the biodiversity of the gastric microbiota in AAG patients was significantly lower in males than in females. It was reported that reduced biodiversity is strictly connected with dysbiosis which, in turn, was proposed as a predisposing factor for neoplastic degeneration, mainly through influencing the host cell proliferation and death, altering immune system activity, and affecting host metabolism [26]. Gunathilake et al. reported that the gastric microbiota were less complex, with lower biodiversity in women affected by gastric cancer compared to healthy controls. Indeed, in men, the dysbiosis index was similar between gastric cancer cases and controls [24]. These contrasting results could be explained by the different settings taken into account; the cited study included gastric cancer patients, while in our study, patients with AAG, a preneoplastic condition, were included. Moreover, different dietary habits, lifestyles and genetic backgrounds between Western and Eastern populations are conceivable.

Even if AAG is epidemiologically more frequent among women [10,27], gastric cancer, a possible long-term complication of AAG, is more common in men. Gastric cancer is reported as the fifth most common cancer in the general population, and men have a nearly two-fold higher incidence rate than women (men: 15.8; women 7.0/100.000) [28]. Perhaps, amongst other factors, the stronger impact of the non-acidic intragastric environment on male patients, with AAG reducing biodiversity, might be another co-player in the higher frequency of gastric cancer in males. 

Moreover, our results showed that in AAG patients at the genus level, *Gemella*, *Haemophilus*, and *Neisseria* were significantly more frequent in women than in men. A study on the human gastric microbiota in hypochlorhydric states reported that at the genus level, after *Streptococcus*, *Gemella* and *Haemophilus* were the most represented genera in the AAG patients [15], which was confirmed also in another study [13], even if gender stratification was not performed. AAG occurs more frequently in women, which is similar to other autoimmune diseases. The higher presence of these bacterial phyla in female AAG patients raises the question of whether they could have a potential etiopathogenetic role in the development of AAG.

Among the main limitations of this study, we are aware of the relatively small sample number and the specific lack of data on the diet and smoking habits of the study population. Moreover, gastroscopy does not represent a physiological setting, for example, because of several hours of fasting and dehydration that are necessary for the exam’s execution but that can modify the dynamism of the gastric microbiota. Finally, the gastric microbiota were assessed by DNA sequencing, a technique that cannot discriminate live from dead bacteria, nor resident from transient bacterial flora. 

## 5. Conclusions

In conclusion, the current study shows that the gender differences in gastric microbiota, particularly a higher gastric bacterial abundance and less microbial diversity in healthy women compared to men, seem to be reset in autoimmune atrophic gastritis, likely due to hypochlorhydria and the non-acid intragastric environment, reducing biodiversity to a greater extent in men. The impact of these findings on gastric physiology and the gender-specific natural history of autoimmune atrophic gastritis remains to be elucidated; in any case, gender differences deserve attention in gastric microbiota studies. For sure, further studies on intragastric microbiota, taking into account gender differences, are welcome to point out not only the effective role of the microbiota as a potential determinant of diseases, but also the gender predisposition to the disease.

## Figures and Tables

**Figure 1 microorganisms-11-01938-f001:**
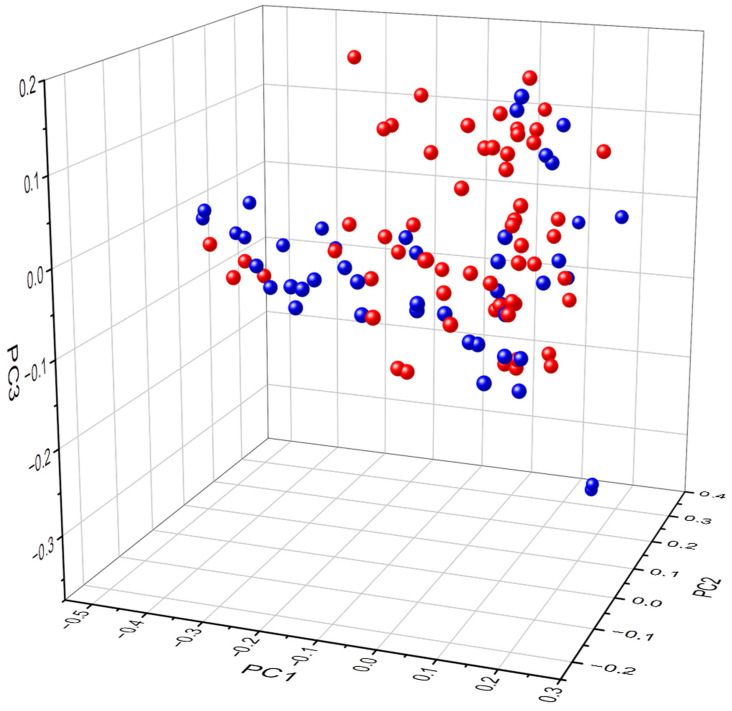
3D Principal Coordinate Analysis of beta-diversity of bacterial genera abundance based on Bray–Curtis matrix of subjects with healthy stomachs and patients with autoimmune atrophic gastritis subdivided by gender (red spheres = females, blu spheres = males, *p* < 0.05). *x*,*y*,*z*-axis PC1, PC2, PC3.

**Figure 2 microorganisms-11-01938-f002:**
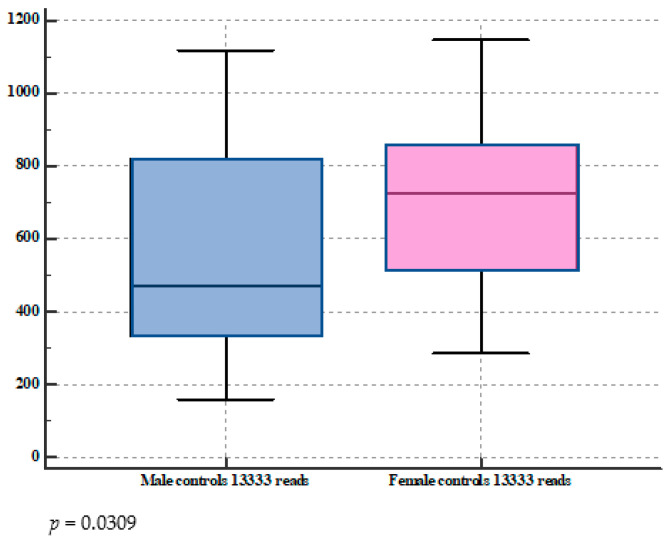
Box-and-whisker graphs comparing OTUs at 13.333 reads between female and male subjects with healthy stomachs (**top**) and autoimmune atrophic gastritis (**bottom**).

**Figure 3 microorganisms-11-01938-f003:**
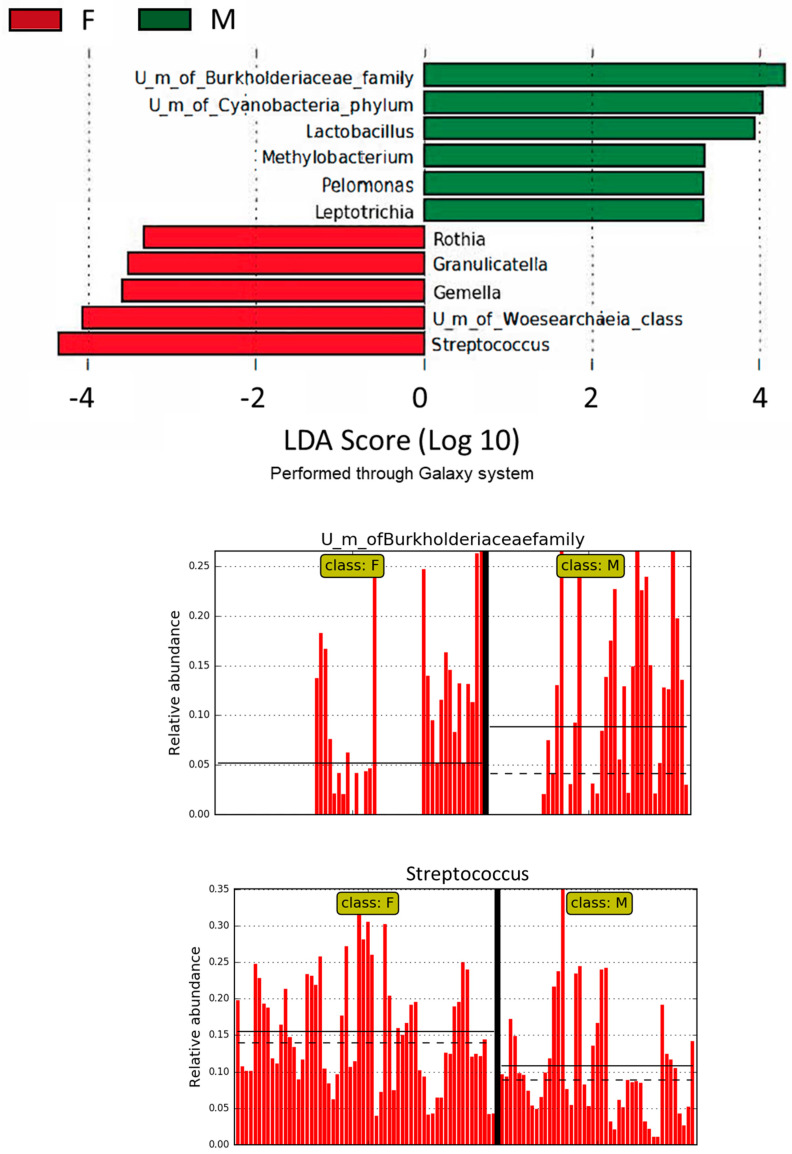
LEfSe analysis based on the average relative abundance of all OTUs with >50% prevalence in female (F) or male (M) subgroups at the genus level according to their LDA score/effect size); mean (solid line) ± SE (dashed line).

**Table 1 microorganisms-11-01938-t001:** Main clinical characteristics of the included subjects with healthy stomachs and autoimmune atrophic gastritis (AAG).

	Subjects with Healthy Stomachs *n* = 30 (57.7%)	Patients with AAG *n* = 22 (42.3%)	*p*
Females/males, *n* (%)	15 (50)/15 (50)	17 (77.3)/5 (22.7)	Ns
Age, years, median (range)	58 (18–83)	64 (36–81)	0.02
Positivity to parietal cells antibodies	0	17 (73.9)	<0.001
Severe corpus atrophy	Na	14 (60.9%)	

Data are expressed as numbers (%), except for age (median, range). Na = not applicable. Ns = not significant.

**Table 2 microorganisms-11-01938-t002:** Comparison of OTUs at different reads between male and female subjects with healthy stomachs and autoimmune atrophic gastritis.

Subjects with Healthy Stomachs			
Reads	Females	Males	*p*
30,000	867.20 (476.10–1365.60)	709.40 (243.60–1250.60)	0.05
26,666	835.20 (474.10–1334.00)	639.20 (239.70–1240.30)	0.04
23,333	796.40 (287.00–1308.30)	533.85 (162.00–1220.80)	0.06
20,000	766.60 (286.60–1264.50)	486.20 (161.80–1196.90)	0.04
16,667	752.40 (285.80–1213.30)	485.00 (161.50–1172.40)	0.04
13,333	704.65 (284.10–1145.00)	470.40 (159.60–1116.60)	0.03
10,000	668.00 (279.40–1062.90)	452.20 (156.40–1051.20)	0.04
6667	619.65 (269.60–921.00)	438.40 (149.00–936.70)	0.05
Patients with autoimmune atrophic gastritis			
Reads	Females	Males	p
30.000	503.15 (263.80–1251.30)	524.50 (415.20–611.00)	0.89
26,666	498.85 (263.60–1228.00)	519.50 (366.80–603.70)	0.80
23,333	487.10 (260.90–1198.80)	512.70 (365.50–598.00)	0.94
20,000	482.25 (257.90–1152.70)	482.45 (203.00–587.70)	0.54
16,667	476.85 (256.40–1108.60)	447.60 (161.00–578. 40)	0.27
13,333	467.20 (174.60–1042.20)	433.00 (160.60–560.50)	0.32
10,000	456.30 (173.00–953.90)	391.05 (140.00–539.70)	0.14
6667	421.25 (169.60–837.30)	361.70 (139.00–501.60)	0.10

**Table 3 microorganisms-11-01938-t003:** Comparison of the Shannon index (alpha diversity) between male and female subjects with healthy stomachs and autoimmune atrophic gastritis.

	Females	Males	*p*-Value
Healthy stomachs	3169	3414	0.003
Autoimmune atrophic gastritis	3505	2345	<0.0001
*p-value*	*p = 0.002*	*p < 0.001*	

**Table 4 microorganisms-11-01938-t004:** Comparison of taxa frequency at the phylum level between female and male subjects with healthy stomachs and autoimmune atrophic gastritis (AAG).

Taxonomy Phyla	HEALTHY STOMACHS n = 30		*p*-Value	AAG n = 22		*p*-Value
	Females Median	Males Median		Females Median	Males Median	
Firmicutes	22.67	15.83	0.5677	27.46	33.11	0.0155
Proteobacteria	18.70	26.41	0.8489	19.13	19.32	0.1425
Bacteroidetes	10.55	10.46	0.1824	5.17	2.36	0.4042
Actinobacteria	2.97	1.75	0.6424	2.19	2.10	0.0042
Fusobacteria	0.19	1.31	0.0124	0.58	0.18	0.4634

**Table 5 microorganisms-11-01938-t005:** Comparison of taxa frequency at the genus level between female and male subjects with healthy stomachs and autoimmune atrophic gastritis (AAG).

Taxonomy Genus	HEALTHY STOMACHS *n* = 30		*p*-Value		AAG *n* = 22		*p*-Value
	Females Mean	Males Mean			Females Median	Males Median	
*Streptococcus*	11.26	7.88	0.0358	*Streptococcus*	15.49	15.94	0.7751
*Prevotella 7*	3.24	2.23	0.0269	*Haemophilus*	2.81	0.59	0.0223
*Neisseria*	1.59	1.36	0.5675	*Prevotella 7*	1.21	0.85	0.4899
*Granulicatella*	1.03	0.73	0.0020	*Neisseria*	1.17	0.43	0.0156
				*Gemella*	1.16	0.37	0.0057

## Data Availability

Data are available upon specific request.

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
