# Peer review of "Gastric Microbiota Gender Differences in Subjects with Healthy Stomachs and Autoimmune Atrophic Gastritis"

_microorganisms, 2023, doi:10.3390/microorganisms11081938_

Round 1
Reviewer 1 Report (Previous Reviewer 3)
The authors have revised the manuscript. However, I still have three suggestions to further improve the manuscript:
1, In figure 2, the title of the Y axis is missing. The authors must revise.
2, Table 3 and Table 5 should be in a three-line format. The authors must revise.
Author Response
Comments and Suggestions for Authors
The authors have revised the manuscript. However, I still have three suggestions to further improve the manuscript:
1, In figure 2, the title of the Y axis is missing. The authors must revise.
As stated in the figure legend, the x,y,z-axis are PC1, PC2, PC3.
2, Table 3 and Table 5 should be in a three-line format. The authors must revise.
As suggested by the Reviewer, the table has been revised in a three-line format.
Reviewer 2 Report (New Reviewer)
This study conducted a post hoc analysis of 52 subjects with either healthy stomachs or autoimmune atrophic gastritis to assess gender differences in gastric bacterial microbiota. Gastric biopsies were obtained for histopathology and DNA extraction, and the microbial composition was assessed using 16S rRNA gene sequencing. Results showed that women with healthy stomachs had higher bacterial abundance and lower microbial diversity compared to men. In autoimmune atrophic gastritis, gender differences were reset, with reduced biodiversity observed in males. These findings highlight the importance of gender in gastric microbiota studies and suggest potential implications for the natural history of autoimmune atrophic gastritis. This manuscript was well-written and structured, I only have minor comments.
Line 112. The version of QIIME software suite?
Line 115-116. Please double-check if it is Operational Taxonomic Units (OUT) or amplicon sequence variant (ASV).
Table 5. It would be better to use graph to present these data.
Figure 3 (lower panel) is difficulty to understand, please use Figure 2 format to present your data.
Author Response
Comments and Suggestions for Authors
This study conducted a post hoc analysis of 52 subjects with either healthy stomachs or autoimmune atrophic gastritis to assess gender differences in gastric bacterial microbiota. Gastric biopsies were obtained for histopathology and DNA extraction, and the microbial composition was assessed using 16S rRNA gene sequencing. Results showed that women with healthy stomachs had higher bacterial abundance and lower microbial diversity compared to men. In autoimmune atrophic gastritis, gender differences were reset, with reduced biodiversity observed in males. These findings highlight the importance of gender in gastric microbiota studies and suggest potential implications for the natural history of autoimmune atrophic gastritis. This manuscript was well-written and structured, I only have minor comments.
Line 112. The version of QIIME software suite?
Thanks, we apologize for this mistake. The version is QIIME 2 and has now been correctly stated in the manuscript.
Line 115-116. Please double-check if it is Operational Taxonomic Units (OUT) or amplicon sequence variant (ASV).
Thanks, It is operational taxonomic units.
Table 5. It would be better to use graph to present these data.
We thank for this suggestion, but unfortunately our manuscript has only four Tables. Which one is meant? If the Reviewer meant Table 4, we tried to construct a graph with these data, but we found it was not as clear as the Table, so we preferred to include the Table in the manuscript.
Figure 3 (lower panel) is difficulty to understand, please use Figure 2 format to present your data.
Thanks for the suggestion, LEfSe analysis was requested in the former revision, and the output of the statistical software has this format. If the Editors agree, we can eventually delete the lower part of the figure.
This manuscript is a resubmission of an earlier submission. The following is a list of the peer review reports and author responses from that submission.
Round 1
Reviewer 1 Report
The article entitled “Gastric microbiota gender differences in subjects with healthy 2 stomachs and autoimmune atrophic gastritis” assessed gender differences in gastric bacterial microbiota between subjects with healthy stomachs 16 and autoimmune atrophic gastritis. Major concerns are shown below.
1. There are 30 healthy subjects and 22 AAG patients. The sample size is small especially when the analysis was performed by genders. Considering the great variabilities in human gut microbiome, gender-specific differences obtained in current study were not convincible.
2. Why did the authors only analyze the gender differences in health subjects and AAG patients other than comparing cases and controls? This should be included in the study.
3. The authors should include baseline characteristics analysis on all the enrolled subjects in the paper.
4. Tables and figures are of low quality. For example, axis titles of fig 1 are vague.
5. Typos should be corrected throughout the paper. For example, “6SrRNA gene” should be corrected as “16SrRNA gene” in Line 20.
Reviewer 2 Report
The present study was designed to assess gender differences in gastric microbiota between healthy subjects and autoimmune atrophic gastritis (AAG) patients. The rationale of this is the already known effect of gender in the prevalence, severity, and natural history of autoimmune diseases, such as AAG. As it turned out, the results shows that there are consistent differences between genders in the healthy stomach microbiota, namely that women have a higher gastric bacterial abundance and less microbial diversity; and that this difference is obscured or not existing in AAG subjects.
In my view, the paper is very interesting, and also very original, since no similar work exist in the literature. Unfortunately, the causes of the gender differences are not clear, and therefore totally hypothetical. Personally, I am not in favour of using the term "dysbiosis" to describe the reduction of biodiversity in AAG patients, but the discussion is otherwise complete and updated.
Reviewer 3 Report
The manuscript by Giulia Pivetta et al. investigated the gender differences of the gastric microbiota in subjects with healthy stomachs and autoimmune atrophic gastritis. The study is of interest to the readers and I have the following comments and suggestions:
1, Table 4 should be in a three-line format.
2, Lefse analysis and Heatmap analysis of the gut microbiota should be added. The analysis of the gut microbiota data is too preliminary in the present study.
3, Figure 1 should be revised. I can hardly read the labelings.
4, Line 265-266, P. gingivalis, T. forsythia, and T. denticola should be italic. The authors must carefully revise the whole manuscript.